# Object Goal Navigation using
# Goal-Oriented Semantic Exploration

**Devendra Singh Chaplot**[1][†]**, Dhiraj Gandhi**[2]**, Abhinav Gupta**[1,2]*****, Ruslan Salakhutdinov**[1]*****
[1]Carnegie Mellon University, [2]Facebook AI Research

Project webpage: https://devendrachaplot.github.io/projects/semantic-exploration
Code: https://github.com/devendrachaplot/Object-Goal-Navigation

## Abstract

This work studies the problem of object goal navigation which involves navigating to an instance of the given object category in unseen environments. End-to-end learning-based navigation methods struggle at this task as they are ineffective at exploration and long-term planning. We propose a modular system called, 'Goal-Oriented Semantic Exploration' which builds an episodic semantic map and uses it to explore the environment efficiently based on the goal object category. Empirical results in visually realistic simulation environments show that the proposed model outperforms a wide range of baselines including end-to-end learning-based methods as well as modular map-based methods and led to the winning entry of the CVPR-2020 Habitat ObjectNav Challenge. Ablation analysis indicates that the proposed model learns semantic priors of the relative arrangement of objects in a scene, and uses them to explore efficiently. Domain-agnostic module design allows us to transfer our model to a mobile robot platform and achieve similar performance for object goal navigation in the real-world.

## 1 Introduction

Autonomous navigation is a core requirement in building intelligent embodied agents. Consider an autonomous agent being asked to navigate to a 'dining table' in an unseen environment as shown in Figure 1. In terms of semantic understanding, this task not only involves object detection, i.e. what does a 'dining table' look like, but also scene understanding of where 'dining tables' are more likely to be found. The latter requires a long-term episodic memory as well as learning semantic priors on the relative arrangement of objects in a scene. Long-term episodic memory allows the agent to keep track of explored and unexplored areas. Learning semantic priors allows the agent to also use the episodic memory to decide which region to explore next in order to find the target object in the least amount of time.

How do we design a computational model for building an episodic memory and using it effectively based on semantic priors for efficient navigation in unseen environments? One popular approach is to use end-to-end reinforcement or imitation learning with recurrent neural networks to build episodic memory and learn semantic priors implicitly [31, 17, 50, 32]. However, end-to-end learning-based methods suffer from large sample complexity and poor generalization as they memorize object locations and appearance in training environments.

Recently, Chaplot et al. [10] introduced a modular learning-based system called 'Active Neural SLAM' which builds explicit obstacle maps to maintain episodic memory. Explicit maps also allow analytical path planning and thus lead to significantly better exploration and sample complexity. However, Active Neural SLAM, designed for maximizing exploration coverage, does not encode

---

[†]Correspondence: chaplot@cs.cmu.edu

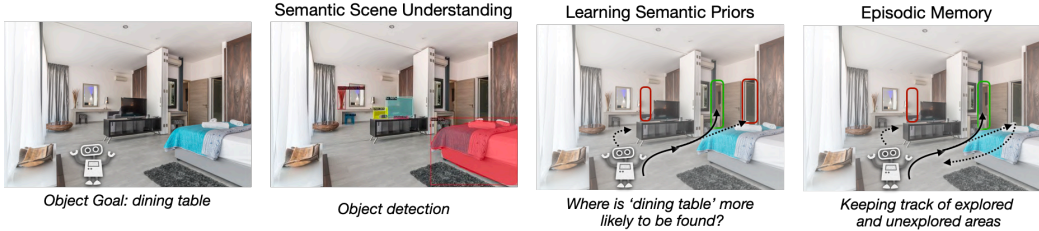

Figure 1: **Semantic Skills required for Object Goal navigation.** Efficient Object Goal navigation not only requires passive skills such as object detection, but also active skills such as an building an episodic memory and using it effective to learn semantic priors abour relative arrangements of objects in a scene.

semantics in the episodic memory and thus does not learn semantic priors. In this paper, we extend the Active Neural SLAM system to build explicit semantic maps and learn semantic priors using a semantically-aware long-term policy.

The proposed method, called 'Goal-Oriented Semantic Exploration' (SemExp), makes two improvements over [10] to tackle semantic navigation tasks. First, it builds top-down metric maps similar to [10] but adds extra channels to encode semantic categories explicitly. Instead of predicting the top-down maps directly from the first-person image as in [10], we use first-person predictions followed by differentiable geometric projections. This allows us to leverage existing pretrained object detection and semantic segmentation models to build semantic maps instead of learning from scratch. Second, instead of using a coverage maximizing goal-agnostic exploration policy based only on obstacle maps, we train a goal-oriented semantic exploration policy which learns semantic priors for efficient navigation. These improvements allow us to tackle a challenging object goal navigation task. Our experiments in visually realistic simulation environments show that SemExp outperforms prior methods by a significant margin. The proposed model also won the CVPR 2020 Habitat ObjectNav Challenge [3][3]. We also demonstrate that SemExp achieves similar performance in the real-world when transferred to a mobile robot platform.

## 2    Related Work

We briefly discuss related work on semantic mapping and navigation below.

**Semantic Mapping.** There's a large body of work on building obstacle maps both in 2D and 3D using structure from motion and Simultaneous Localization and Mapping (SLAM) [21, 23, 42]. We defer the interested readers to the survey by Fuentes-Pacheco et al. [14] on SLAM. Some of the more relevant works incorporate semantics in the map using probabilistic graphical models [4] or using recent learning-based computer vision models [49, 30]. In contrast to these works, we use differentiable projection operations to learn semantic mapping with supervision in the map space. This limits large errors in the map due to small errors in first-person semantic predictions.

**Navigation.** Classical navigation approaches use explicit geometric maps to compute paths to goal locations via path planning [24, 27, 5, 41]. The goals are selected base on heuristics such as the Frontier-based Exploration algorithm [47]. In contrast, we use a learning-based policy to use semantic priors for selecting exploration goals based on the object goal category.

Recent learning-based approaches use end-to-end reinforcement or imitation learning for training navigation policies. These include methods which use recurrent neural networks [31, 26, 7, 38, 22, 8, 39, 43], structured spatial representations [17, 34, 9, 20, 16] and topological representations [36, 37]. Recent works tackling object goal navigation include [45, 48, 44, 32]. Wu et al. [45] try to explore structural similarities between the environment by building a probabilistic graphical model over the semantic information like room types. Similarly, Yang et al. [48] propose to incorporate semantic priors into a deep reinforcement learning framework by using Graph Convolutional Networks. Wortsman et al. [44] propose a meta-reinforcement learning approach where an agent learns a self-supervised interaction loss that encourages effective navigation to even keep learning in a test environment. Mousavian et al. [32] use semantic segmentation and detection masks obtained by running state-of-the-art computer vision algorithms on the input observation and used a deep network to learn the navigation policy based on it. In all the above methods, the learnt representations are implicit and the models need to learn obstacle avoidance, episodic memory, planning as well

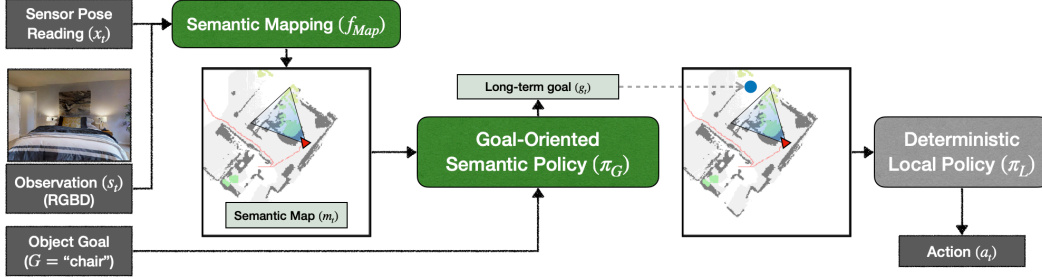

Figure 2: **Goal-Oriented Semantic Exploration Model Overview.** The proposed model consists of two modules, Semantic Mapping and Goal-Oriented Semantic Policy. The Semantic Mapping model builds a semantic map over time and the Goal-Oriented Semantic Policy selects a long-term goal based on the semantic map to reach the given object goal efficiently. A deterministic local policy based on analytical planners is used to take low-level navigation actions to reach the long-term goal.

as semantic priors implicitly from the goal-driven reward. Explicit map representation has been shown to improve performance as well as sample efficiency over end-to-end learning-based methods for different navigation tasks [10, 12], however they learn semantics implicitly. In this work, we use explicit structured semantic map representation, which allows us to learn semantically-aware exploration policies and tackle the object-goal navigation task. Concurrent work studies the use of similar semantic maps in learning exploration policies for improving object detection systems [11].

## 3 Method

**Object Goal Task Definition.** In the Object Goal task [38, 1], the objective is to navigate to an instance of the given object category such as 'chair' or 'bed'. The agent is initialized at a random location in the environment and receives the goal object category ($G$) as input. At each time step $t$, the agent receives visual observations ($s_t$) and sensor pose readings $x_t$ and takes navigational actions $a_t$. The visual observations consist of first-person RGB and depth images. The action space $\mathcal{A}$ consists of four actions: `move_forward`, `turn_left`, `turn_right`, `stop`. The agent needs to take the 'stop' action when it believes it has reached close to the goal object. If the distance to the goal object is less than some threshold, $d_s(= 1m)$, when the agent takes the stop action, the episode is considered successful. The episode terminates at after a fixed maximum number of timesteps ($= 500$).

**Overview.** We propose a modular model called 'Goal-Oriented Semantic Exploration' (SemExp) to tackle the Object Goal navigation task (see Figure 2 for an overview). It consists of two learnable modules, 'Semantic Mapping' and 'Goal-Oriented Semantic Policy'. The Semantic Mapping module builds a semantic map over time and the Goal-Oriented Semantic Policy selects a long-term goal based on the semantic map to reach the given object goal efficiently. A deterministic local policy based on analytical planners is used to take low-level navigation actions to reach the long-term goal. We first describe the semantic map representation used by our model and then describe the modules.

**Semantic Map Representation.** The SemExp model internally maintains a semantic metric map, $m_t$ and pose of the agent $x_t$. The spatial map, $m_t$, is a $K \times M \times M$ matrix where $M \times M$ denotes the map size and each element in this spatial map corresponds to a cell of size $25cm^2$ ($5cm \times 5cm$) in the physical world. $K = C + 2$ is the number of channels in the semantic map, where $C$ is the total number of semantic categories. The first two channels represent obstacles and explored area and the rest of the channels each represent an object category. Each element in a channel represents whether the corresponding location is an obstacle, explored, or contains an object of the corresponding category. The map is initialized with all zeros at the beginning of an episode, $m_0 = [0]^{K \times M \times M}$. The pose $x_t \in \mathbb{R}^3$ denotes the $x$ and $y$ coordinates of the agent and the orientation of the agent at time $t$. The agent always starts at the center of the map facing east at the beginning of the episode, $x_0 = (M/2, M/2, 0.0)$.

**Semantic Mapping.** In order to a build semantic map, we need to predict semantic categories and segmentation of the objects seen in visual observations. It is desirable to use existing object detection and semantic segmentation models instead of learning from scratch. The Active Neural SLAM model predicts the top-down map directly from RGB observations and thus, does not have any mechanism for incorporating pretrained object detection or semantic segmentation systems.

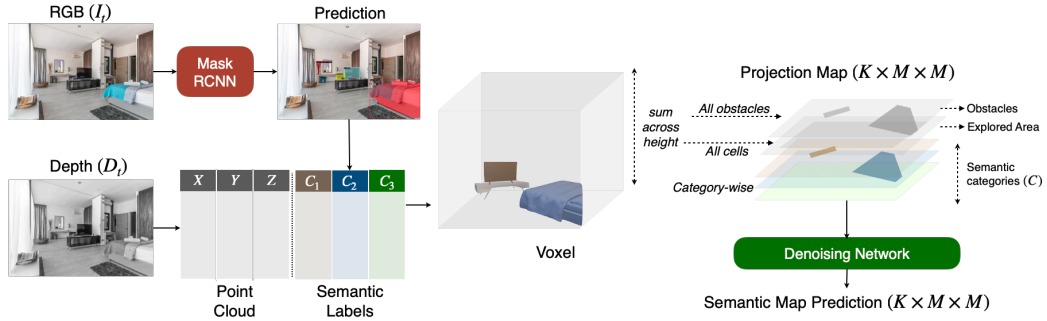

Figure 3: **Semantic Mapping.** The Semantic Mapping module takes in a sequence of RGB ($I_t$) and Depth ($D_t$) images and produces a top-down Semantic Map.

Instead, we predict semantic segmentation in the first-person view and use differentiable projection to transform first-person predictions to top-down maps. This allows us to use existing pretrained models for first-person semantic segmentation. However small errors in first-person semantic segmentation can lead to large errors in the map after projection. We overcome this limitation by imposing a loss in the map space in addition to the first-person space.

Figure 3 shows an overview of the Semantic Mapping module. The depth observation is used to compute a point cloud. Each point in the point cloud is associated with the predicted semantic categories. The semantic categories are predicted using a pretrained Mask RCNN [18] on the RGB observation. Each point in the point cloud is then projected in 3D space using differentiable geometric computations to get the voxel representation. The voxel representation is then converted to the semantic map. Summing over the height dimension of the voxel representation for all obstacles, all cells, and each category gives different channels of the projected semantic map. The projected semantic map is then passed through a denoising neural network to get the final semantic map prediction. The map is aggregated over time using spatial transformations and channel-wise pooling as described in [10]. The Semantic Mapping module is trained using supervised learning with cross-entropy loss on the semantic segmentation as well as semantic map prediction. The geometric projection is implemented using differentiable operations such that the loss on the semantic map prediction can be backpropagated through the entire module if desired.

**Goal-Oriented Semantic Policy.** The Goal-Oriented Semantic Policy decides a long-term goal based on the current semantic map to reach the given object goal ($G$). If the channel corresponding to category $G$ has a non-zero element, meaning that the object goal is observed, it simply selects all non-zero elements as the long-term goal. If the object goal is not observed, the Goal-Oriented Semantic Policy needs to select a long-term goal where a goal category object is most likely to be found. This requires learning semantic priors on the relative arrangement of objects and areas. We use a neural network to learn these semantic priors. It takes the semantic map, the agent's current and past locations, and the object goal as input and predicts a long-term goal in the top-down map space.

The Goal-Oriented Semantic Policy is trained using reinforcement learning with distance reduced to the nearest goal object as the reward. We sample the long-term goal at a coarse time-scale, once every $u = 25$ steps, similar to the goal-agnostic Global Policy in [10]. This reduces the time-horizon for exploration in RL exponentially and consequently, reduces the sample complexity.

**Deterministic Local Policy.** The local policy uses Fast Marching Method [41] to plan a path to the long-term goal from the current location based on the obstacle channel of the semantic map. It simply takes deterministic actions along the path to reach the long-term goal. We use a deterministic local policy as compared to a trained local policy in [10] as they led to a similar performance in our experiments. Note that although the above Semantic Policy acts at a coarse time scale, the Local Policy acts at a fine time scale. At each time step, we update the map and replan the path to the long-term goal.

## 4   Experimental Setup

We use the Gibson [46] and Matterport3D (MP3D) [6] datasets in the Habitat simulator [39] for our experiments. Both Gibson and MP3D consist of scenes which are 3D reconstructions of real-world environments. For the Gibson dataset,wWe use the train and val splits of Gibson tiny set for training

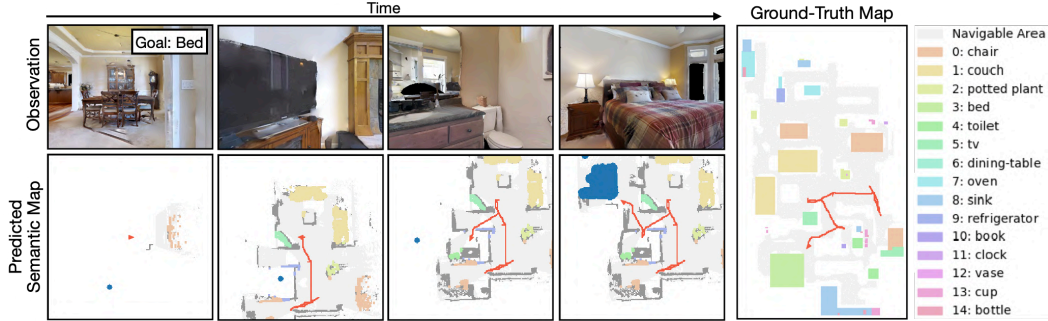

Figure 4: **Example Trajectory.** Figure showing an example trajectory of the SemExp model in a scene from the Gibson test set. Sample images seen by the agent are shown on the top and the predicted semantic map is shown below. The goal object is 'bed'. The long-term goal selected by the Goal-driven Semantic Policy is shown in blue. The ground-truth map (not visible to the agent) with the agent trajectory is shown on the right for reference.

and testing respectively as the test set is held-out for the online evaluation server. We do not use the validation set for hyper-parameter tuning. The semantic annotations for the Gibson tiny set are available from Armeni et al. [2]. For the MP3D dataset, we use the standard train and test splits. Our training and test set consists of a total of 86 scenes (25 Gibson tiny and 61 MP3D) and 16 scenes (5 Gibson tiny and 11 MP3D), respectively.

The observation space consists of RGBD images of size $4 \times 640 \times 480$ , base odometry sensor readings of size $3 \times 1$ denoting the change in agent's x-y coordinates and orientation, and goal object category represented as an integer. The actions space consists of four actions: move_forward, turn_left, turn_right, stop. The success threshold $d_s$ is set to $1m$. The maximum episode length is 500 steps. We note that the depth and pose are perfect in simulation, but these challenges are orthogonal to the focus of this paper and prior works have shown that both can be estimated effectively from RGB images and noisy sensor pose readings [15, 10]. These design choices are identical to the CVPR 2020 Object Goal Navigation Challenge.

For the object goal, we use object categories which are common between Gibson, MP3D, and MS-COCO [28] datasets. This leads to set of 6 object goal categories: 'chair', 'couch', 'potted plant', 'bed', 'toilet' and 'tv'. We use a Mask-RCNN [18] using Feature Pyramid Networks [29] with ResNet50 [19] backbone pretrained on MS-COCO for object detection and instance segmentation. Although we use 6 categories for object goals, we build a semantic map with 15 categories (shown on the right in Figure 4) to encode more information for learning semantic priors.

**Architecture and Hyperparameter details.** We use PyTorch [35] for implementing and training our model. The denoising network in the Semantic Mapping module is a 5-layer fully convolutional network. We freeze the Mask RCNN weights in the Semantic Mapping module (except for results on Habitat Challenge in Section 5.2) as Matterport does not contain labels for all 15 categories in our semantic map. We train the denoising network with the map-based loss on all 15 categories for Gibson frames and only 6 categories on MP3D frames. The Goal-driven Semantic Policy is a 5 layer convolutional network followed by 3 fully connected layers. In addition to the semantic map, we also pass the agent orientation and goal object category index as separate inputs to this Policy. They are processed by separate Embedding layers and added as an input to the fully-connected layers.

We train both the modules with 86 parallel threads, with each thread using one scene in the training set. We maintain a FIFO memory of size 500000 for training the Semantic Mapping module. After one step in each thread, we perform 10 updates to the Semantic Mapping module with a batch size of 64. We use Adam optimizer with a learning rate of $0.0001$. We use binary cross-entropy loss for semantic map prediction. The Goal-driven Policy samples a new goal every $u = 25$ timesteps. For training this policy, we use Proximal Policy Optimization (PPO) [40] with a time horizon of 20 steps, 36 mini-batches, and 4 epochs in each PPO update. Our PPO implementation is based on [25]. The reward for the policy is the decrease in distance to the nearest goal object. We use Adam optimizer with a learning rate of $0.000025$, a discount factor of $\gamma = 0.99$, an entropy coefficient of $0.001$, value loss coefficient of $0.5$ for training the Goal-driven Policy.

Table 1: **Results.** Performance of SemExp as compared to the baselines on the Gibson and MP3D datasets across three different metrics: Success rate, SPL [1] (Success weighted by Path Length) and DTS (Distance To Goal in meters). See text for more details.

| Method | Gibson | | | MP3D | | |
|---|---|---|---|---|---|---|
| | SPL | Success | DTS (m) | SPL | Success | DTS (m) |
| Random | 0.004 | 0.004 | 3.893 | 0.005 | 0.005 | 8.048 |
| RGBD + RL [39] | 0.027 | 0.082 | 3.310 | 0.017 | 0.037 | 7.654 |
| RGBD + Semantics + RL [32] | 0.049 | 0.159 | 3.203 | 0.015 | 0.031 | 7.612 |
| Classical Map + FBE [47] | 0.124 | 0.403 | 2.432 | 0.117 | 0.311 | 7.102 |
| Active Neural SLAM [10] | 0.145 | 0.446 | 2.275 | 0.119 | 0.321 | 7.056 |
| SemExp | **0.199** | **0.544** | **1.723** | **0.144** | **0.360** | **6.733** |

**Metrics.** We use 3 metrics for comparing all the methods: **Success.** Ratio of episodes where the method was successful. **SPL.** Success weighted by Path Length as proposed by [1]. This metric measures the efficiency of reaching the goal in addition to the success rate. **DTS:** Distance to Success. This is the distance of the agent from the success threshold boundary when the episode ends. This is computed as follows:

$$DTS = max(||x_T - G||_2 - d_s, 0)$$

where $||x_T - G||_2$ is the L2 distance of the agent from the goal location at the end of the episode, $d_s$ is the success threshold.

## 4.1 Baselines.

We use two end-to-end Reinforcement Learning (RL) methods as baselines:

**RGBD + RL:** A vanilla recurrent RL Policy initialized with ResNet18 [19] backbone followed by a GRU adapted from Savva et al. [39]. Agent pose and goal object category are passed through an embedding layer and append to the recurrent layer input.

**RGBD + Semantics + RL [32]:** This baseline is adapted from Mousavian et al. [32] who pass semantic segmentation and object detection predictions along with RGBD input to a recurrent RL policy. We use a pretrained Mask RCNN identical to the one used in the proposed model for semantic segmentation and object detection in this baseline. RGBD observations are encoded with a ResNet18 backbone visual encoder, and agent pose and goal object are encoded usinng embedding layers as described above.

Both the RL based baselines are trained with Proximal Policy Optimization [40] using a dense reward of distance reduced to the nearest goal object. We design two more baselines based on goal-agnostic exploration methods combined with heuristic-based local goal-driven policy.

**Classical Mapping + FBE [47]:** This baseline use classical robotics pipeline for mapping followed by classical frontier-based exploration (FBE) [47] algorithm. We use a heuristic-based local policy using a pretrained Mask-RCNN. Whenever the Mask RCNN detects the goal object category, the local policy tries to go towards the object using an analytical planner.

**Active Neural SLAM [10]:** In this baseline, we use an exploration policy trained to maximize coverage from [10], followed by the heuristic-based local policy as described above.

## 5 Results

We train all the baselines and the proposed model for 10 million frames and evaluate them on the Gibson and MP3D scenes in our test set separately. We run 200 evaluations episode per scene, leading to a total of 1000 episodes in Gibson (5 scenes) and 2000 episodes in MP3D (10 scenes, 1 scene did not contain any object of the 6 possible categories). Figure 4 visualizes an exmaple trajectory using the proposed SemExp showing the agent observations and predicted semantic map[4]. The quantitative results are shown in Table 1. SemExp outperforms all the baselines by a considerable margin consistently across both the datasets (achieving a success rate 54.4%/36.0% on Gibson/MP3D vs 44.6%/32.1% for the Active Neural SLAM baseline) . The absolute numbers are higher on the Gibson set, as the scenes are comparatively smaller. The Distance to Success (DTS) threshold for Random in Table 1 indicates the difficulty of the dataset. Interestingly, the baseline combining classical exploration with pretrained object detectors outperforms the end-to-end RL baselines. We observed that the training performance of the RL-based baselines was much higher indicating that

Table 2: **Ablations and Error Analysis.** Table showing comparison of the proposed model, SemExp, with 2 ablations and with Ground Truth semantic segmentation on the Gibson dataset.

| Method | SPL | Success | DTS (m) |
|---|---|---|---|
| SemExp w.o. Semantic Map | 0.165 | 0.488 | 2.084 |
| SemExp w.o. Goal Policy | 0.148 | 0.450 | 2.315 |
| SemExp | 0.199 | 0.544 | 1.723 |
| SemExp w. GT SemSeg | 0.457 | 0.731 | 1.089 |

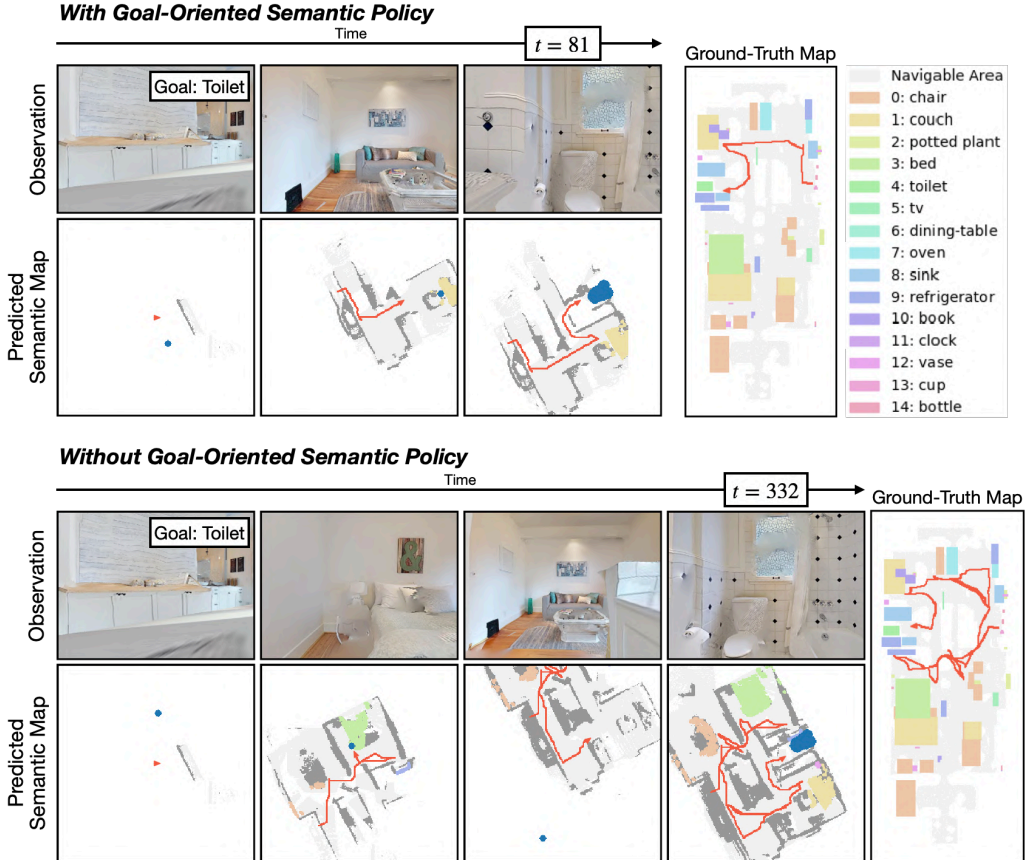

Figure 5: Figure showing an example comparing the proposed model with (**top**) and without (**bottom**) Goal-Oriented Semantic Policy. Starting at the same location with the same goal object of 'toilet', the proposed model with Goal-Oriented Policy can find the target object much faster than without Goal-Oriented Exploration.

they memorize the object locations and appearance in the training scenes and generalize poorly. The increase in performance of SemExp over the Active Neural SLAM baseline shows the importance of incorporating semantics and the goal object in exploration.

## 5.1 Ablations and Error Analysis

To understand the importance of both the modules in SemExp, we consider two ablations:

**SemExp w.o. Semantic Map.** We replace the Semantic Map with the Obstacle-only Map. As opposed to the Active Neural SLAM baseline, the Goal-oriented Policy is still trained with distance reduced to the nearest object as the reward.

**SemExp w.o. Goal Policy.** We replace the Goal-driven policy with a goal-agnostic policy trained to maximize exploration coverage as in [10], but still trained with the semantic map as input.

The results in the top part of Table 2 show the performance of these ablations. The performance of SemExp without the Goal-oriented policy is comparable to the Active Neural SLAM baseline, indicating that the Goal-oriented policy learns semantic priors for better exploration leading to

Table 3: **Results on CVPR 2020 Habitat ObjectNav Challenge.** Table showing the performance of top 5 entries on the Test-Challenge dataset. Our submission based on the SemExp model won the challenge.

| Team Name | SPL | Success | Dist To Goal (m) |
|---|---|---|---|
| **SemExp** | **0.102** | **0.253** | **6.328** |
| SRCB-robot-sudoer | 0.099 | 0.188 | 6.908 |
| Active Exploration (Pre-explore) | 0.046 | 0.126 | 7.336 |
| Black Sheep | 0.028 | 0.101 | 7.033 |
| Blue Ox | 0.021 | 0.069 | 7.233 |

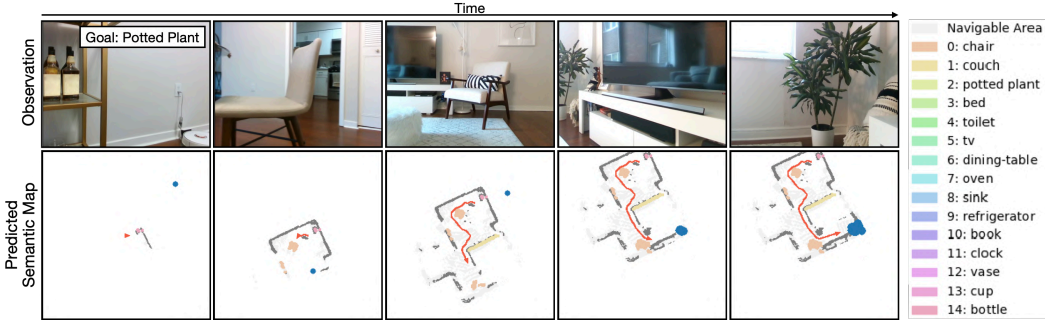

Figure 6: **Real-world Transfer.** Figure showing an example trajectory of the SemExp model transferred to the real-world for the object goal 'potted plant'. Sample images seen by the robot are shown on the top and the predicted semantic map is shown below. The long-term goal selected by the Goal-driven Policy is shown in blue.

more efficient navigation. Figure 5 shows an qualitative example indicating the importance of the Goal-oriented Policy. The performance of SemExp without the Semantic Map also drops, but it is higher than the ablation without Goal Policy. This indicates that it is possible to learn some semantic priors with just the obstacle map without semantic channels.

The performance of the proposed model is still far from perfect. We would like to understand the error modes for future improvements. We observed two main sources of errors, semantic segmentation inaccuracies and inability to find the goal object. In order to quantify the effect of both the error modes, we evaluate the proposed model with ground truth semantic segmentation (see **SemExp w. GT SemSeg** in Table 2) using the 'Semantic' sensor in Habitat simulator. This leads to a success rate of 73.1% vs 54.4%, which means around 19% performance can be improved with better semantic segmentation. The rest of the 27% episodes are mostly cases where the goal object is not found, which can be improved with better semantic exploration.

## 5.2 Result on Habitat Challenge

SemExp also won the CVPR 2020 Habitat ObjectNav Challenge. The challenge task setup is identical to ours except the goal object categories. The challenge uses 21 object categories for the goal object. As these object categories are not overlapping with the COCO categories, we use DeepLabv3 [13] model for semantic segmentation. We predict the semantic segmentation and the semantic map with all 40 categories in the MP3D dataset including the 21 goal categories. We fine-tune the DeepLabv3 segmentation model by retraining the final layer to predict semantic segmentation for all 40 categories. This segmentation model is also trained with the map-based loss in addition to the first-person segmentation loss. The performance of the top 5 entries to the challenge are shown in Table 3. The proposed approach outperforms all the other entries with a success rate of 25.3% as compared to 18.8% for the second place entry.

## 5.3 Real World Transfer

We used the Locobot hardware platform and PyRobot API [33] to deploy the trained policy in the real world. In Figure 6 we show an episode of the robot when it was provided 'potted plant' as the object goal. The long-term goals sampled by the Goal-driven policy (shown by blue circles on the map) are often towards spaces where there are high chances of finding a potted plant. This indicates that it is learning to exploit the structure in the semantic map to some extent. Out of 20 trials in the real-world, our method succeeded in 13 episodes leading to a success rate of 65%. End-to-end learning-based policies failed consistently in the real-world due to the visual domain gap between

simulation environments and the real-world. Our model performs well in the real-world as (1) it is able to leverage Mask RCNN which is trained on real-world data and (2) the other learned modules (map denoising and the goal policy) work on top-down maps which are domain-agnostic. Our trials in the real-world also indicate that perfect pose and depth are not critical to the success of our model as it can be successfully transferred to the real-world where pose and depth are noisy.

# 6    Conclusion

In this paper, we presented a semantically-aware exploration model to tackle the object goal navigation task in large realistic environments. The proposed model makes two major improvements over prior methods, incorporating semantics in explicit episodic memory and learning goal-oriented semantic exploration policies. Our method achieves state-of-the-art performance on the object goal navigation task and won the CVPR2020 Habitat ObjectNav challenge. Ablation studies show that the proposed model learns semantic priors which lead to more efficient goal-driven navigation. Domain-agnostic module design led to successful transfer of our model to the real-world. We also analyze the error modes for our model and quantify the scope for improvement along two important dimensions (semantic mapping and goal-oriented exploration) in the future work. The proposed model can also be extended to tackle a sequence of object goals by utilizing the episodic map for more efficient navigation for subsequent goals.

# 7    Broader Impact Statement

As discussed in the paper, this work has the potential to enable robots to better exploit the structure in the real world we live in to do better object goal navigation. This will be a quintessential criterion when mobile robots with manipulators will be ubiquitous helping elders or those with disabilities with their day to day life chores.

There could be privacy concerns as these robots work on the visual feed taken in everyday settings. However, we may be able to bypass the issue if we anonymize the data. In addition to that we have also shown policy learned only in simulation has potential to get transferred to the real world. If this trained policy works reliably in diverse settings then we might not even have to record any data.

## Acknowledgements

Carnegie Mellon University effort was supported in part by the US Army W911NF1920104, IARPA D17PC00340 and ONR Grant N000141812861. Devendra Singh Chaplot and Ruslan Salakhutdinov would also like to acknowledge NVIDIA's GPU support.

**Licenses for referenced datasets:**
Gibson: http://svl.stanford.edu/gibson2/assets/GDS_agreement.pdf
Matterport3D: http://kaldir.vc.in.tum.de/matterport/MP_TOS.pdf,
https://niessner.github.io/Matterport/

## Footnotes

[3]https://aihabitat.org/challenge/2020/

[4]See demo videos at https://devendrachaplot.github.io/projects/semantic-exploration

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
