[Supplementary Material]

# Supplementary Material: Object Goal Navigation using Semantically Aware Exploration

## A  Visualizations

Figure 1: Figure showing an example comparing the proposed model with (**top**) and without (**bottom**) Goal-Oriented Semantic Exploration. Starting at the same location with the same goal object of 'toilet', the proposed model with Goal-Oriented Exploration can find the target object much faster than without Goal-Oriented Exploration.

# B  Hyperparameter and Architecture Details

We use PyTorch [3] for implementing and training our model. The denoising network in the Semantic Mapping module is a 5-layer fully convolutional network. We freeze the Mask RCNN weights in the Semantic Mapping module (except for results on Habitat Challenge) as Matterport does not contain labels for all 15 categories in our semantic map. We train the denoising network with the map-based loss on all 15 categories for Gibson frames and only 6 categories on MP3D frames.

The Goal-Oriented Semantic Policy is a 5 layer convolutional network followed by 3 fully connected layers. The PyTorch code for the architecture of 5 convolutional layers is the following:

```
self.conv_layers = nn.Sequential(
    nn.MaxPool2d(2),
    nn.Conv2d(23, 32, 3, stride=1, padding=1),
    nn.ReLU(),
    nn.MaxPool2d(2),
    nn.Conv2d(32, 64, 3, stride=1, padding=1),
    nn.ReLU(),
    nn.MaxPool2d(2),
    nn.Conv2d(64, 128, 3, stride=1, padding=1),
    nn.ReLU(),
    nn.MaxPool2d(2),
    nn.Conv2d(128, 64, 3, stride=1, padding=1),
    nn.ReLU(),
    nn.Conv2d(64, 32, 3, stride=1, padding=1),
    nn.ReLU(),
    Flatten()
)
```

We use local and global maps similar to [1]. The 23 channel input to the Goal-Oriented Semantic Policy policy consists of the following: 4 channels for local map prediction (obstacles, explored area, current agent location, past agent locations), 4 channels for global map prediction (resized to local map size), 15 channels of local semantic category predictions.

In addition to the semantic map, we also pass the agent orientation and goal object category index as separate inputs to this Policy. They are processed by separate Embedding layers of size 8 each and added as an input to the fully-connected layers. The first fully connected layers map the convolutional layer output with embeddings to a size of 256. The input and output size of next 2 fully connected layers is 256, followed by value and action predictions for reinforcement learning.

We train both the modules with 86 parallel threads, with each thread using one scene in the training set. We maintain a FIFO memory of size 500000 for training the Semantic Mapping module. After one step in each thread, we perform 10 updates to the Semantic Mapping module with a batch size of 64. We use Adam optimizer with a learning rate of 0.0001. We use binary cross-entropy loss for semantic map prediction. The Goal-Oriented Semantic Policy samples a new goal every $u = 25$ timesteps. For training this policy, we use Proximal Policy Optimization (PPO) [4] with a time horizon of 20 steps, 36 mini-batches, and 4 epochs in each PPO update. Our PPO implementation is based on [2]. The reward for the policy is the decrease in distance to the nearest goal object. We use Adam optimizer with a learning rate of 0.000025, a discount factor of $\gamma = 0.99$, an entropy coefficient of 0.001, value loss coefficient of 0.5 for training the Goal-Oriented Semantic Policy. We will open-source the code for training and evaluating our model.

# C  Object categories

The 15 object categories used in the semantic map were: chair, couch, potted plant, bed, toilet, tv, dining-table, oven, sink, refrigerator, book, clock, vase, cup, bottle. Out of these, the first 6 (chair, couch, potted plant, bed, toilet, tv) are categories common between Gibson, MP3D and MS-COCO datasets are used for object goals.