[Reviews · NeurIPS 2020]

Review 1

Summary and Contributions: The work aims to solve the problem of locating objects in unknown environments in a navigation task. It combines a semantic SLAM approach (mapping the environment using predetermined object categories) with a medium-term planner that learns object priors (i.e. TVs tend to be in living rooms, not kitchens).

Strengths: This is one of the best papers I've read this NeurIPS. The goal is clear, the story is clear, the writing is neat, the experiments are great, meaningful baselines are all there, even a real robot experiment, ablations are provided for the core method. And most importantly, the method itself is a clear algorithmic innovation (over Active Neural SLAM) that will have significant relevance in the embodied navigation community. On top of all that, the method was "battle-tested" in a hard navigation challenge with a held-out test set and scored first place in terms of success rate at the time of writing.

Weaknesses: 1) The work is made significantly harder to reproduce by omitting implementation details of the 3D projection "using differentiable geometric computations". The fact that those are used is mentioned multiple times on page 4 but it's entirely unclear to me what that entails. It would've been nice to at least include a rough sketch of the code. 2) Similarly, the order of pretrainings and trainings is hard to parse and I would've appreciated a block of pseudocode of the training procedure or at least an ordered list of training steps. UPDATE: It would've been nice to release code already with the submission as opposed to promising to do it in the future and the reviewers having to trust the authors blindly. In the past, even when the authors promised code release with the acceptance, it still took several months for a release. I'd like to encourage earlier and reliable code release with _submission_ instead of acceptance.

Correctness: The method is fundamentally sound. The experimental setup is meaningful wrt to the method and the results are sufficiently discussed.

Clarity: The paper is very well written.

Relation to Prior Work: The paper identifies Active Neural SLAM as most related and clearly distinguishes itself from that prior work. Other navigation works are mentioned in sufficient detail.

Reproducibility: Yes

Additional Feedback: Minor nitpicks: - Section 1, line 19/20, "keep a track" -> "keep track" - Section 3, line 84, "and take navigational actions" -> "and takes ..." - How exactly does that "summing over height dimension" work? Let's say there's a TV on a shelf, with shelving above and below the TV - wouldn't the TV get lost when that's just projected down? Or rather how do you disentangle the different semantic maps from the summed up representation? - Is the denoising network pretrained? - In Fig 4 and in the suppl. video, it would've been nice if you rotate either the agent's map or the GT map to fit the other for viewer convenience - (very IMHO) In Tab 1 it would've been nice to include a very brief mention of "SPL" and "DTS" in the caption - How'd you manage for the paragraph between line 210 and 211 to not have line numbers? There's a typo in there, last sentence "this computed as follows" -> "this is computed..." - As mentioned above, huge kudos for adding a real robot experiment. However, I feel like you're slightly undercutting your own results here by not giving us the random baselines results. Without this or any information about the house, the reader cannot judge the difficulty of the task and the relative benefit of this method. - The Broader Impact section is also one of the better ones I've read this NeurIPS. I appreciate that you kept to concrete problems and didn't entirely spitball potential future societal impacts. However, it would've been nice to still develop this a little further. For example, do you think that this technology is also capable of locating non-elderly/non-disabled humans? Are there technical problems in having a robot navigate human spaces (see e.g. Anca Dragans works in that area)? And is this something that you could see in a production setting a few years down the line or do you think real-life problems are usually better solved with fixed maps and classic navigation?


Review 2

Summary and Contributions: The paper introduces a system for goal-driven, semantic navigation called semantically aware exploration. It builds upon neural slam to maintain an explicit semantic map of an environment given a point cloud and semantic labels that are projected onto the map. The map is then used to find a semantic goal. If the goal is already in the map, then the location is selected and the a path is planned via FMM. If the goal is not in the map, a goal selection policy that is RL trained is used to use semantic information is used. The results are demonstrated on several data sets and ablated over.

Strengths: The paper is interesting and well written. The combination of learning to leverage both data availability (the semantic projection) and the nuanced relationships of semantic information for goal selection along with classical methods for planning and maintaining an overall map is reasonable. The results are promising, showing significant gains compared to other methods as well as showing the benefits of each decision. The demonstrated experiments are realistic and the authors include an on robot experiment. The paper adds several new features beyond [6], none of which are hugely novel, but together they are an interesting and informative case study. The video is nice to see.

Weaknesses: There are a few areas within the paper that could be better explained or should be investigated further: > The denoising network and semantic projection is not well defined. It is stated that “We overcome this limitation by imposing a loss in the map space in addition to the first-person space.”. What is the loss in first-person space, I understood the semantic mask to be fully pretrained. How does the differentiable geometric computation work exactly. How does the denoised semantic map merge with the current map? How does this perform compared to performance without the denoising network? > A human baseline on semantic exploration would put the results in better performance context. > SemSeg experiments are mentioned for having a 19% improvement in success rate, but they have a very significant improvement in SPL. What is the cause of this? Only small regions of the goal being visible? > Where does the system fail most often? Does it get stuck in local minimum, choose bad explorations, etc. > Please quantify “We observed that the training performance of the RL-based baselines was much 223 higher indicating that they memorize the object locations and appearance in the training scenes and 224 generalize poorly.” === After rebuttal === After reading the rebuttal, I plan to keep my score the same. Overall, I believe they achieve SOTA with an interesting method on a difficult problem, though there are improvements and more detailed evaluations that would improve the work.

Correctness: The claims are well backed up with ablations and the empirical results are reasonably thorough.

Clarity: The paper is well written. Though a few aspects should be more clearly stated in the main body as discussed in weaknesses.

Relation to Prior Work: The relation to prior work is reasonable.

Reproducibility: Yes

Additional Feedback: A few areas that would be interesting to see: > How does the RL trained goal prediction use semantic information and how often is it in error? The former could be studied by, given a map and varying semantic object targets, where is the goal placed? > How would this work if the agent were given multiple targets successively in the same environment? It would be interesting to see the curve of information gain effecting SPL for each target. > [6] Considers noise in the motion and sensor data, how would this effect your results?


Review 3

Summary and Contributions: This work proposes an explicit semantic mapping representation combined with a long-horizon waypoint-based policy supported by a hard-coded motion controller to tackle the problem of navigating to a target object category in a previously unseen environment. The paper's proposed contributions include top-down semantic mapping, high-level waypoint-based navigation, and demonstration of the results on a real robot.

Strengths: This is a well-motivated and challenging task, and the solution proposed is reasonable. The semantic segmentation loss term in top-down mapping space appears novel, if minor. The demonstration on a real robot is valuable and definitely elevates the significance of the claims.

Weaknesses: The novelty of the proposed approach is fairly low. Top-down local mapping via projection of a depth-based sensor is a standard approach here, as is semantic segmentation as the representation of choice. Long-horizon waypoint-based navigation implemented by a hard-coded low-level navigation policy is also a standard technique.

Correctness: The claims and method appear to be correct; the empirical methodology is good, and the inclusion of a real robot puts this work above many of the other concurrent pieces of work that take otherwise very similar approaches to this problem.

Clarity: The paper is well-written, the problem is well motivated and important, and the approach is justified.

Relation to Prior Work: Prior work is discussed, although the novelty over previous and concurrent work is not thoroughly established.

Reproducibility: Yes

Additional Feedback: I think this general approach is reasonable and unfortunately somewhat standard; the specific contributions here of including a loss term in the projected map space and including a real robot evaluation, while important and good, do not in my view justify publication at this time. ---- UPDATE ---- After considering the authors' response, and discussion with the other reviewers, I have decided to revise my score to a 6. While certain metrics still have very far to go (6 meter average distance to goal, hardly better than random), the success rate of SOTA is still very low, and this is a worthy contribution that improves SOTA on a very hard generalization task.


Review 4

Summary and Contributions: This paper presents an extension to recent work on Active Neural SLAM [1], where semantic information about object categories is explicitly incorporated into the model. The extensions in the model architecture provide explicit semantic information about the various objects of the scene in the generated 2D map, that allows an agent to navigate in its environment and find a specified goal object much efficiently compared to baselines. Some of these baselines use - and others do not - semantic information. The comparison was performed using Gibson [2] and Matterport3D (MP3D) [3], which include 3D reconstructions of real environments. Training was performed on 86 scenes and testing on 16. There are 6 possible goal object categories (but a total of 15 categories for all objects in the scenes) and 4 possible actions. Segmentation and classification is performed on the input image using a pre-trained Mask-RNN. A labelled point cloud is then generated, which is then projected into a 2D semantic map. A denoising loss is applied on the semantic map. The semantic map is then used along with the agent’s pose and goal object to predict a long-term goal in the 3D map (implemented as a neural network), which is in turn used to select the action, using a local deterministic policy to plan a sequence of actions (using Fast Marching Method). The long term goal policy is trained using PPO, and is sampled every few timesteps. Importantly, the model has been submitted to the CVPR 2020 Habitat Object Goal Navigation Challenge [4], where it ranks first and second, depending on the split. In this challenge the agent is tasked with navigating to a specified object (e.g. toilet or bed) inside a rendered 3D house environment. [1] Devendra Singh Chaplot, Dhiraj Gandhi, Saurabh Gupta, Abhinav Gupta, and Ruslan Salakhut- dinov. Learning to explore using active neural slam. In International Conference on Learning Representations (ICLR), 2020. [2] Fei Xia, Amir R. Zamir, Zhi-Yang He, Alexander Sax, Jitendra Malik, and Silvio Savarese. Gibson Env: real-world perception for embodied agents. In Computer Vision and Pattern Recognition (CVPR), 2018 IEEE Conference on. IEEE, 2018. [3] Angel Chang, Angela Dai, Thomas Funkhouser, Maciej Halber, Matthias Niebner, Manolis Savva, Shuran Song, Andy Zeng, and Yinda Zhang. Matterport3d: Learning from rgb-d data in indoor environments. In 2017 International Conference on 3D Vision (3DV), pages 667–676. IEEE, 2017. [4] Kadian, A., Truong, J., Gokaslan, A., Clegg, A., Wijmans, E., Lee, S., Savva, M., Chernova, S. and Batra, D., 2019. Are we making real progress in simulated environments? measuring the sim2real gap in embodied visual navigation. arXiv preprint arXiv:1912.06321. Competition: https://evalai.cloudcv.org/web/challenges/challenge-page/580/leaderboard/1634

Strengths: 1. The model performance is state-of-the-art, as shown by its holding of the first and second place (depending on the split) of the CVPR 2020 Habitat Object Goal Navigation Challenge. 2. The authors report that they have implemented the algorithm on a Locobot hardware platform acting in a real environment with promising results. The algorithm is efficient and can run in real time on a robot with limited computational resources. 3. An ablation study is presented, showing the contribution of the different model components to the final performance in the simulated environment. 4. The paper provides important empirical evidence for how maintaining semantic information about objects in an agent’s representation can lead to more efficient exploration in navigation tasks.

Weaknesses: 1. I am a bit concerned about how significantly novel this work is, as it brings together many existing methods. As a final result the algorithm presented in the paper seems to do great, but I am not sure whether it is a significant scientific contribution that can benefit the wider community. To me, it seems to be a well engineered approach that was designed to solve the CVPR 2020 Habitat Object Goal Navigation Challenge. --POST-REBUTTAL--- Having considered the authors' response and reviewer's discussion, I can see that there is value in the specific approach for the embodied indoors navigation community. 2. Evaluation conducted using 102 scenes from the simulated dataset. There is no mention of why or how these scenes were selected, whether they are complex enough, or whether they pose a significant challenge. Will their proposed method scale to any environment? Some statistics about the selected scenes are necessary. How big are the selected scenes? How many rooms? How many of each target object in each scene? List the scenes used in the appendix for reproducibility. --POST-REBUTTAL--- I am happy with the author's response on this. 3. Though great to have reported real world transfer, the experiments run seem to be minimal and the evaluation premature. --POST-REBUTTAL--- I am happy with the author's response on this. 4. There is overall lack of a deeper discussion about the importance and implications of this work for the wider research effort in autonomous agents and how the authors expect the proposed method to scale. In addition, the paper is overly focused on results from the datasets and challenge, with little interpretation of these results. 5. No standard deviations are reported in the quantitative results. ---POST-REBUTTAL--- The authors have not addressed this concern. I strongly believe that results without any notion of deviation or error (without a good reason for their absence) are not reliable. 6. I am not sure if I missed it, but I could not find an explanation of the “Random” baseline. Is this random at the level of the low level action (local policy) or random at the level of the long-term goal? Also, the distance to success values reported in Table 1 are not that much larger than that achieved by the proposed model. This is of course difficult to judge because of the lack of more scene information or results interpretation. --POST-REBUTTAL--- I am happy with the author's response on this. I understand now that this is indeed a challenging task - looking at the very low success rate of baselines.

Correctness: - Line 268-270: “The long-term goals [...] the structure in the semantic map.” - This claim should have been backed up by quantitative evidence. What is shown in Figure 5 is a single image and the statement seems to be overly generalising - this can be an area where a more thorough quantitative analysis can be undertaken by the authors. --POST-REBUTTAL--- I still think there's only a few example from which this seems too strong of a statement to make. - How good is segmentation and classification? From the ablation study it seems that providing the ground truth segmentation leads to a much better navigation performance, but there’s no direct evaluation of the segmentation module per se. If the goal is real world navigation then segmentation and classification should be discussed more. --POST-REBUTTAL--- Happy with author's response. - Line 9-10: “Ablation analysis indicates that the proposed model learns semantic priors of the relative arrangement of objects in a scene, and uses them to explore efficiently.” I am worried that this statement in the abstract is misleading and portraits a wrong conclusion from the ablation study. The ablation study, as far as I understand, shows that if the variables encoding “semantic” information are removed from the map (i.e. only by providing an obstacle-only map instead of the semantic map - I assume by removing the class channels) then the overall performance is impacted negatively. This does not imply that the model “learns semantic priors”, nor can we reach any conclusions about how (the mechanism through which) these representations are used by the model (I am referring to the claim about “the relative arrangements”). The validity of the claim about semantics, for example, could be verified through carefully assessing the accuracy of the segmentation and classification in each scene. --POST-REBUTTAL--- Please see my response in additional feedback section below. Not enough space here.

Clarity: The paper is well written.

Relation to Prior Work: The relation to prior work is clear, but the authors offer no substantial discussion of the impact of this novelty other than mentioning that robotic assistants navigating in the real world would benefit from this algorithm. At the same time, the majority of the results are from the competition and the relevant datasets, while the results from implementing it in the real world are only mentioned in passing. In my view, if the end goal is real world navigation, then ideally we would expect a more thorough evaluation of the progress made by this work compared to previous work, what still remains unsolved, and what should be the next area to focus on in order to overcome these limitations. It would also be interesting to know what the views of the authors are regarding the limitations, if any, originating from using pre-trained segmentation and classification components.

Reproducibility: Yes

Additional Feedback: Even though the results on the two datasets are currently NOT reproducible because the selected scenes for training and testing are not mentioned, I consider the competition entry good-enough for responding "yes" to the reproducibility question above. I strongly believe the information about the specific scenes should be provided by the authors. ---POST-REBUTTAL--- This has been satisfactorily addressed by the authors. I am very enthusiastic about object-level semantics and the approach taken by the authors, and I do think that they clearly illustrate that they found an efficient way to solve object goal navigation in realistic looking environments. On the other hand, it seems that all empirical work, and its presentation, is focused on solving the CVPR 2020 Habitat Object Goal Navigation Challenge and only illustrating its robustness to the real world in a short section without much analysis or discussion. I am not very familiar with this area of research, so I cannot suggest specific analysis that the authors can undertake, but the paper’s value for the community would significantly increase by such an informative analysis - beyond simply benchmarking against baselines - that shows 1) how difficult the problem is and 2) how much progress we can make with this method. One way in which the paper could be improved is by providing more varied examples of environments it can effectively navigate in, to illustrate a wide applicability. Another is by focusing on a particular application in the real world, in order to illustrate that the proposed model should be widely used from sim2real transfer for navigation tasks. Overall, the paper seems to present a very promising model, but the results and analysis are somewhat premature. ---POST-REBUTTAL--- Most of my concerns have been addressed. ---POST-REBUTTAL--- Continued last point in "Correctness" section above: I am happy with the authors' response about the segmentation possibly not being the best way to assess whether the model has learned semantic priors. Still, their response has not helped me see how the original statement in the abstract is correct - and I hence advice to make the statement less strong regarding the efficiency of exploration (the challenge is far from solved).

[Author Response · NeurIPS 2020]

We thank the reviewers for their valuable feedback and comments. We address the concerns below:

**Regarding novelty (R3, R5).** R3 & R5 point out that parts of some modules are based on prior work. Firstly, note that
the complete method is significantly different from prior methods ([25,37,38,41]) tackling the object goal navigation
task. In fact, we propose an alternate paradigm of approaching this problem using learning-based modular models
as opposed to end-to-end learning in prior methods. We show that we can still get the benefits of learning within
the modules, but avoid the drawbacks of high sample complexity and memorization in end-to-end learning. The
effectiveness of modular methods is not evident without our experiments which we believe the NeurIPS community
benefits from knowing.

That said, even among the modules, the use of differentiable semantic mapping and a goal-driven semantic policy on
top-down maps is novel. Since our goal-driven policy and map denoising operate only on top-down maps (which are
domain-invariant) and our semantic mapping leverages MaskRCNN trained on real-world images, our system can be
transferred from sim-to-real. This is very different from end-to-end learning policies which are known to struggle under
domain shift. Novelty is also recognized by R1 ("clear algorithmic innovation") and R2 ("adds several new features").

**More real-world experiments (R1, R5).** All reviewers have appreciated the real-world experiments in the submission.
R1 & R5 have suggested there should be more emphasis on real-world experiments. Note that prior works for this
task typically conduct only simulation experiments and no real-world experiments on a mobile robot platform (for eg.
[25,37,38,41]). Nevertheless, we performed more real-world experiments, testing our model and 2 baselines across
100 episodes each in two different home environments. The proposed model achieved a 62% success rate vs 46% for
Active Neural SLAM [6] and 0% for the RGBD+Semantics+RL [25] baseline (either just going forward and colliding
or turning on the spot for most episodes). The RL baseline reached close to the goal object in 6 episodes, but it did
not take the stop action. These results highlight the difficulty of the task and the significance of our approach. Our
domain-invariant modular design allows real-world transfer while end-to-end RL based baselines suffer from visual
domain shift. Semantic mapping and goal-driven exploration result in improvement over [6]. Note that [6] is not
designed for the object goal task, we adapt it for this task and use it as a strong baseline.

**Regarding datasets and implementation details (R1, R2, R5).** We will open-source complete code for training and
evaluating our model including differentiable geometric computations (R1, R2), order of pretrainings and trainings
(R1), map denoising (R2), dataset details (R5). Regarding datasets (R5), Gibson and MP3D are based on real-world
reconstructions of indoor homes and are commonly used in visual navigation papers (for eg [6, 11, 36]). We use
standard splits from the original datasets. Scenes are fairly large with an average floor area of Gibson and MP3D scenes
being $368.8m^2$ and $517.3m^2$ respectively. We expect our method to perform similarly in any typical home environment.
Furthermore, the method is not specific to homes or specific object categories used in our experiments, and should also
work on other types of environments and objects if trained on them. We will also add all relevant details regarding
datasets and implementation in the appendix including a list of scenes used (R5) and training pseudocode (R1).

**Regarding error modes and analysis (R2, R5).** *GT SemSeg leads to a large improvement in SPL (R2)*: due to small
regions being visible and detecting objects from a greater distance, pretrained MaskRCNN often doesn't detect occluded
or distant objects. *Where does the system fail most often? (R2)* (a) inefficient exploration, (b) not detecting the object
when visible (quantitative numbers in Sec 5.1 and Table 2). We did not observe any local minimum in terms of
exploration coverage (agent getting stuck in an area). This is probably because the explored area is explicitly marked
on the semantic map. *Training performance of RL baselines (R2)*: 0.37 and 0.49 Success for RL and Semantics+RL
in Gibson. *Limitations of using pretrained segmentation (R5)*: Our ablation indicates +19% success when using GT
SemSeg, in practice pretrained MaskRCNN can be tuned to adapt to the environment. *More varied examples (R5)*: we
provide several examples in the supplementary material and video, we will add more examples in future revisions.

**Regarding segmentation and classification (R5).** We do not report segmentation or classification performance as we
do not propose a method for these tasks. Also, reporting segmentation performance in simulation is not very informative
as sim consists of reconstruction errors not present in the real-world. We believe segmentation is not a good indicator of
learning semantic priors either as the segmentation for each frame is independent of each other. The long-term goal
selection is based on the complete episodic map (which includes the relative arrangement of objects across all the
frames), and thus, we believe our ablation is a better indicator of learning semantic priors.

**Other questions. R1:** Summation over height is category-specific. Voxel representation is 4D with 4th channel being
the category. We sum voxels of a particular category for the corresponding semantic map channel. Denoising network
is not pretrained. **R2:** Semantic segmentation is not pretrained in Habitat experiments. Denoised map is merged with
the current map using channel-wise max pooling. Removing denoising leads to a 5% drop in performance. Success in
real-world experiments indicates noisy motion and sensor do not affect our method much. **R5:** Random baseline is
random at low-level action. Regarding performance on the challenge leaderboard, note that our method has the best
performance on both the splits in terms of success rate.

[Meta-Review · NeurIPS 2020]

This paper proposes to train an ObjectNav policy that generalises to unseen environments by using a modular system that classifies objects and builds an episodic semantic map, which it is uses to explore the environment based on the object category, building upon the hierarchical method in "Learning to explore using Active Neural SLAM". The method achieved SOTA performance on the 2020 CVPR Object Goal Navigation Habitat Challenge. Interestingly, the policy, trained on Gibson and MP3D, has been transferred and deployed in a real robot, with some success. While the initial reviews were mixed (9, 7, 4, 5), the reviewers converged on (8, 7, 6, 6), agreeing during discussion that the paper deserved to be accepted. Based on the reviews, I recommend this paper for acceptance as a spotlight or poster presentation. Please note that reviewers R2 and R4 have raised several issues that are not fully addressed by the rebuttal and that should be addressed in the final version (in particular, standard deviations and discussions about the final position of the agent in relatively small houses). As a side note, the authors did a commendable job of acknowledging privacy concerns in datasets based on peoples' homes. As reviewer R4 suggests, and while additional ethical concerns about bumping into humans are rarely addressed in the robot community, mentioning this would be a good addition. I would also disagree with R1's request to release the code at submission rather than after acceptance - this requirement typically penalises large projects, and promotes the release of functional but poorly written and badly documented code (as the experiments are barely wrapped up) rather than code that effectively helps the scientific community. Code release is significant work in itself.